# Introducing Geotourism to Diversify the Visitor Experience in Protected Areas and Reduce Impacts on Overused Attractions

**Daminda Sumanapala** [1,2,*] and **Isabelle D. Wolf** [3,4]

1   Tourism Research Cluster, Curtin University, Perth, WA 6150, Australia
2   College of Arts, Business, Law and Social Sciences, Murdoch University, Murdoch, WA 6150, Australia
3   Australian Centre for Culture, Environment, Society and Space, School of Geography and Sustainable Communities, University of Wollongong, Wollongong, NSW 2522, Australia
4   Centre for Ecosystem Science, University of New South Wales, Sydney, NSW 2052, Australia
*   Correspondence: daminda.sumanapala@murdoch.edu.au

**Abstract:** Sri Lankan National Parks are highly popular among international and local visitors, as they offer close-up encounters with large animal species. Yala National Park is one of the top five parks in the country attracting larger crowds than any other parks especially during the holiday season. Most visitors flock to the park to observe the highly sought-after Asian Elephant and Asian Leopard. This has led to safari operators pursuing these animals aggressively to satisfy visitor expectations, thereby threatening wildlife populations. In this article, we present a straightforward methodology to identify high-potential geotourism sites in order to diversify visitor experiences as a means to alleviate pressure from wildlife tourism. To identify sites, firstly this study has evaluated various place characteristics important for the development of geotourism, including scientific, tourism and 'additional' value indicators. As a result, three sites out of four were selected to promote geotourism in Yala National Park. Secondly, a strengths, weaknesses, opportunities, and threats (SWOT) analysis was performed, which builds on the results from the numerical evaluation but provides a more in-depth narrative assessment. Ultimately, this study serves as an example of how to seize the opportunities that geotourism offers for diversifying tourism offers in protected areas.

**Keywords:** geotourism; wildlife tourism; visitor impacts; diversification; protected area

## 1. Introduction

Geotourism is a developing nature-based tourism branch with great potential to promote and educate on geological and geomorphological landscape attributes of nature [1,2]. The core intention of geotourism experiences is to educate visitors about geological features and processes [3,4], while conserving them for future generations by providing economic benefits to local communities [5,6] and designing activities that are low impact [7–10]. Eder and Patzak (2004) posit that geotourism serves as a tool for developing a form of sustainable tourism that is deeply connected with the natural resources of rocks, minerals, fossils, soils, landforms, and landscapes. The demand for observing and learning about geological features certainly exists [11,12]. Galvao et al. (2022) noted that geotourism has been implemented as a strategy for developing tourism destinations and promoting tourist attractions while preserving geological heritage and popularizing the knowledge of geology [13]. Geotourism was thus coined as a "smart specialization strategy" [14]. Here, we posit that geotourism is also a "smart diversification strategy" that can help expand existing tourism offers to protect overused attractions.

A main attraction in protected areas is wildlife and many visitors come to seek close observations of eye-catching species. To achieve visitor satisfaction, a fine balance needs to be struck between protecting wildlife and providing satisfying experiences that meet expectations by a large number of visitors [15]. Visitor satisfaction from wildlife tourism is strongly based on the immediate experience while traveling [16,17], which has tangible and

intangible attributes such as being emotional, physical, spiritual, and intellectual. These findings corroborate evidence from a recent study by Stoleriu et al. (2019) that revealed that visitors are willing to accept tangible services and facilities during nature-based activities as well as intangible ones [18]. Therefore, Du Preez and Elizabeth (2019) argued that a satisfying experience could be delivered based on geological and geomorphological place features as it both delivers on tangible and intangible attributes of the experience [19–22]. The geotourism experience can be more sustainable if features are chosen that are more resistant to the immediate impact of visitation that is reported for wildlife tourism such as the stress response or avoidance behaviour of sensitive wildlife [23,24].

Furthermore, Gordon and Baker [25] noted that geotourism serves as a bridge that connects people with natural and cultural landscape elements. As a result of that, Aquino et al. [26] argued that through geotourism authentic tourist experiences can be designed that elicit deep interest for learning about the geomorphology of protected areas [27]. In fact, previous studies have confirmed that nature-based visitors are highly educated and motivated to gain knowledge of the country or place of their visit and show concern about local environments [28,29]. To capitalise on this potential, geotourism experiences need to incorporate the "Three G's", namely, geohistory, geointerpretation, and geoconservation [30]. If visitor expectations are met with an adequate combination of interpretation and other elements of the experience the potential of geotourism is great, as confirmed in a recent case study in Turkey [31]. Geotourism experiences are also easily coupled with other immersive recreational activities such as walking, taking photographs, and sightseeing, all of which can add to the learning experience.

Studies have confirmed that diversification is a valid means to reduce the pressure on key visitor attractions, however, this type of research is still quite limited. A notable exception is Moyle et al. [32] who tested alternative visitor experience scenarios to reduce visitation and pressures on the iconic summit of Mount Warning, Australia. The authors [32] recommended that park managers should take necessary action to minimize pressure on overused attractions by creating multi-experience sites or developing multiple sites catering to differing needs. This is the lens through which we view geotourism in this research, namely, as a means to diversify existing visitor experience offers for the benefit of reducing pressure on overused attractions.

Since, the 1990s geotourism has developed widely as a form of nature-based tourism although this trend is lagging behind in the developing countries in South Asia [20]. The Ruban [33] study has shown that geotourism practices and research have not matured in South Asian countries to extent they have elsewhere, for instance in Ethiopia [34], Iran [35], Thailand [36], or India [37]. Recent studies have explored the geotourism opportunities in developing countries such as Morocco, Ecuador, and Jordan [38,39] to explore economic potential and to minimize tourism impacts at selected sites [38]. A study in Spain focussed specifically on introducing geotourism to diversify leisure activities [40].

However, there are some challenges for promoting geotourism as is the case for other nature-based tourism experiences [24,29,41]. Tourism staged in nature-based settings such as protected areas, can have visitor impacts, especially without proper management plans [42]. These impacts might affect wildlife and its habitat through trampling of vegetation, creating undefined tracks and trails, and the construction and maintenance of new trails [43–45]. Wolf and Croft [43] highlighted visitor movements can impact animal behaviour both short and long term [46–48]. Similarly, poorly managed geotourism can have significant impacts in vulnerable and fragile settings [49]. Sumanapala and Wolf [47] have reviewed geotourism impacts focussed on Asia geoparks which revealed two main types of impact: primary impacts from visitor activities relating to for instance erosion and waste disposal, and secondary impacts originating from a lack of policies or legislative frameworks that protect geotourism sites. All these need to be considered for geotourism development to be sustainable.

This paper focuses on identifying suitable geotourism sites in Yala National Park, a highly popular protected area in Sri Lanka. The intention is to showcase how to diversify

visitor experiences within the park as a means to alleviate pressure from overrun wildlife attractions that suffer from visitor impacts. To evaluate geotourism potential, this study has focused on four sites in particular which we will introduce in more detail in the next section. We used pre-defined selection criteria to numerically quantify the potential of each site. Through a subsequent strengths, weaknesses, opportunities, and threats (SWOT) analysis we provide a more in-depth narrative assessment of the development of geotourism at Yala National Park as a solution to minimize the impacts of overuse.

## 2. Methods

### 2.1. Study Area

Sri Lanka is a tropical island in the Indian Ocean with a land area of 65,610 km$^2$ that is characterized by a complex geomorphology [50,51]. The geological basement of Sri Lanka is composed of highly metamorphosed Precambrian rocks which are subdivided into three main lithotectonic units: the Highland Complex, the Wanni Complex (formerly Vijayn Complex), and the Kadugannawa Complex. As for the geomorphology, the country is divided into three main ranges: the coastal lowlands (0–270 m), the uplands (270–1060 m), and the highlands (1060–2240 m) [52]. Although the island is small it has an abundance of geological and geomorphological features: mountain ranges, valleys, flat plains, and what is referred to as the "isolated hills" or Inselbergs which are particularly scenic features of Sri Lanka. In addition, Chandrajith [51] noted that the northern and northwestern coastal stretch underlined by limestone presents isolated igneous intrusions within the Precambrion complex.

Sri Lanka which is one of the most famous tourism destinations in South Asia. Over time, the Sri Lankan tourism industry has undergone a transformation directed away from the 80's sun, sand, and sea image to that of a nature-based tourism mecca. As a result, visitors flock to Sri Lanka to watch wildlife in their natural habitat, especially in wildlife parks, which are protected under the Sri Lankan wildlife ordinance. Some parks are particularly popular such as the Yala, Minneria, and Udawawe National Parks [53,54].

Among these, Yala National Park (Figure 1) stands out as the top park in the country attracting overly large crowds both on- and off season [54,55]. Yala National Park is located in the Southern coastal area of Sri Lanka. In 1938 it was declared as the Ruhuna (now Yala) National Park. It was extended to cover 151,177.8 ha of land, and is home to important fauna and flora characterized by semi-arid thorn scrub and pockets of dense forest [56]. Yala National Park encompasses five blocks (Figure 1). The east side of the national park is a designated "Strict Natural Reserve" [55,57,58] Climatically, Yala is located in a dry zone. The annual rainfall mainly occurs between November to January influenced by the northeast monsoon.

The attractions in Yala National Park are diverse ranging from wildlife to archaeological and geological sites: Yala National Park is most famous though for its wildlife watching opportunities. The main attraction is wildlife, in particular the Asian Leopard (*Panthera pardus kotiya*) and the Asian Elephant (*Elephas maximus*) as well as bird watching as the park is a major stop for migrating birds. In addition, it is famous for its archaeological sites (e.g., Sithulpawwa Rajamaha Viharaya) dating back to the 2nd century BC as it belonged to the kingdom of Magama Knidom of Rhuhuna. Yala National Park is surrounded by other major attractions including cultural heritage places or the Bundala bird sanctuary. It is also conveniently located along a famous tourism route in the country.

Geologically, Yala National Park is located in the Vijayan Complex, which mainly presents with soil Reddish Brown Earth (Alfisols) [59], which contains gneisses, gneissic granites, granitic gneisses, granite, augen, gneisses, and migmatite [60]. In addition, small lower-Miocene-shale limestone beds overlined by Quaternary deposits are present for instance, at Minihagalkanda. Ancient man-made tanks established during the Magama kingdom period provide further geological points of interest [61].

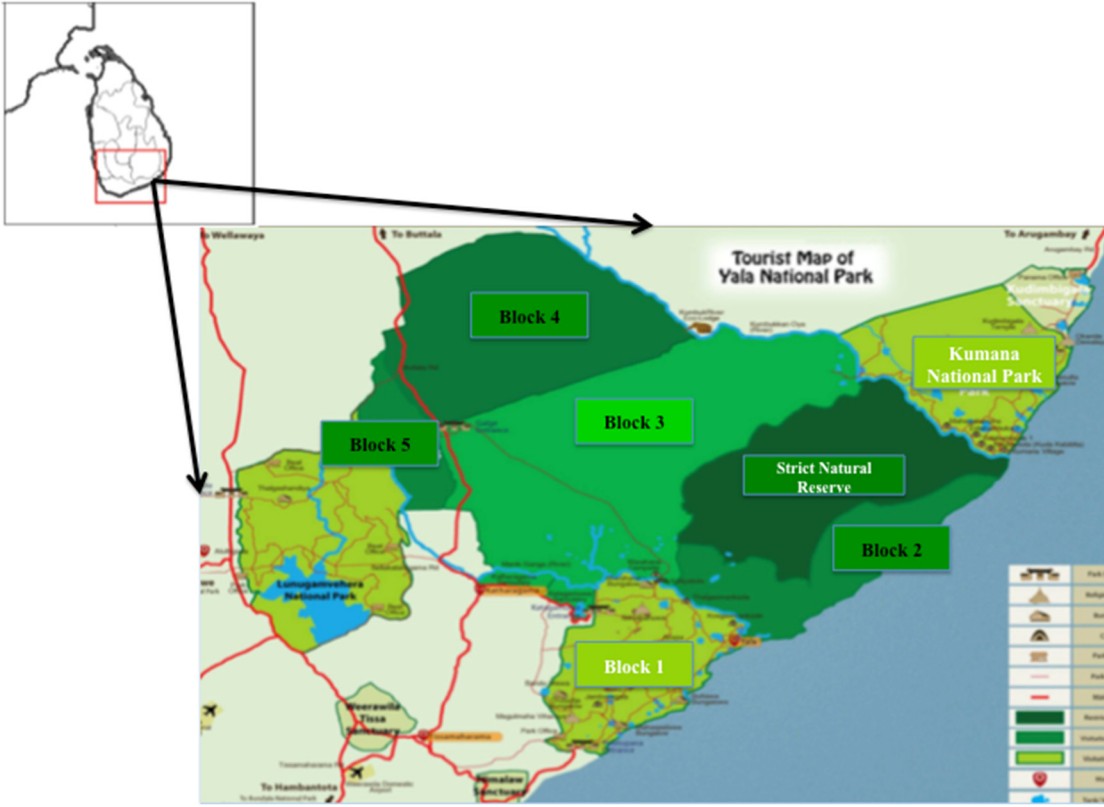

**Figure 1.** A tourist map of Yala National Park (adapted from the Sri Lankan Wildlife Department). Source: Wildlife conservation department of Sri Lanka.

*2.2. Study Sites*

For the development of geotourism experiences we have identified four potential sites: Patanangala, Elephant Rock, Minihagalkanda, and Potana (Figure 2). The key features of the selected potential geotourism sites are presented in Table 1 including their location, their main attraction, additional attractive features important from a tourism perspective, access and distance from the main entrance.

A couple of these are already known among park visitors, namely, Patanangala and Elephant Rock. However, these two sites are more frequented because of their scenic value and not necessarily because of their geomorphological and geological value. Even park managers and other tourism stakeholders have not yet identified these sites for potential geotourism development, so our research is timely. Patanangala, for instance, is popular among park visitors for the scenic views and coastal habitat. It has been primarily promoted as an Inselberg island attraction. Therefore, visitors are enjoying and experiencing the coastal environment and its appealing views of the Inselberg, mostly without being aware of the geomorphological and geological opportunities available at this site. This area would be particularly interesting for studying erosion, but this potential has not yet been investigated either. The situation is similar for Elephant Rock, which is mainly known for its unusual elephant shape. However, apart from the visual appeal, the rock is of metamorphic origin from the Precambrian era. It now provides an important habitat for vegetation and wildlife including elephants.

Minihagalkanda has the unique appearance of a single vertical rock. This site is indeed recognized by park managers and tourism stakeholders for its geomorphological and geological value. However, in this instance, restrictions and the need for a permission to access the site are hindering the development for geotourism. Some studies have been undertaken in this area about the formation of limestone which consists of a 1.5–2.0 m thick layer of a lower, non-fossiliferous basal bed of ferruginous grit and sandstone. On top of that, brownish and yellowish sand and clay have been deposited. Numerous sites of

historical and archaeological value are located in this area. The name "Minihagalkanda" indicates that this area was once inhabited by the pre-historic man of Sri Lanka of which evidence can be found here. The final study site, Potana, which remains almost completely unstudied, adds value through the geological formation of limestone. All four sites are essentially undervalued from a tourism perspective because park management is not yet capitalizing on promoting these sites for their geomorphological and geological value.

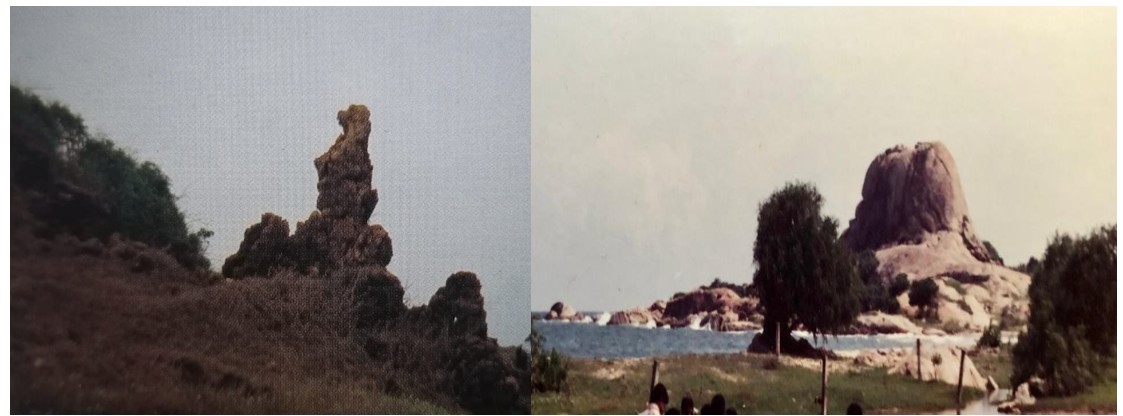

Minihagalkanda
*Photographs Wildlife Department*

Patanangala
*Photographs Upul Wickremasinghe*

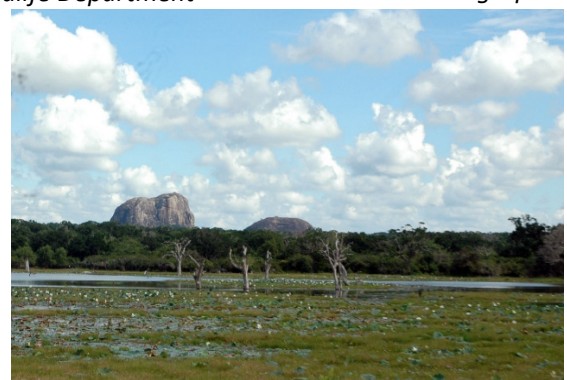

Elephant rock
*Photographs Upul Wickremasinghe*

**Figure 2.** Potential sites for geotourism development in Yala National Park, Sri Lanka.

**Table 1.** Key features of selected geotourism sites at Yala National Park, Sri Lanka.

| Characteristics | Minihagalkanda | Potana | Patanangala | Elephant Rock |
|---|---|---|---|---|
| Location | Block 2 | Block 1 | Block 1 | Block 1 |
| Attraction | Rocks eroded into the shape of a man | Geological formation of limestone | Lagoons and large rocky inselbergs | Rock in the shape of an elephant |
| Important additional features | Multi-coloured sandstone Archeological | Ecological Archeological | Ecological Archeological | Archeological |
| Access | Permission required | No restrictions | No restrictions | No restrictions |
| Distance from main entrance (km) | 60 | 40 | 10 | 15 |

*2.3. Assessment Procedure*

Scholars have applied many different criteria to quantify geotourism potential of sites, which leaves a challenging number of criteria to select from. Zangmo et al. (2017) noted that previous research has adopted the following criteria: scientific value, ecological value, aesthetic value, and cultural value [62–64]. Other studies have also focused on the density of the potential visitor population (e.g., [65]). However, due to the large size of the National Park, the study followed a systematic strategy to collect the primary data about potential geosites. The literature proposes specific steps to identify and assess geosites. Pereria et al. [66] for instance detail six stages for a geosite inventory and assessment including (a) identification of potential geosites, (b) qualitative evaluation, (c) geosite selection, (d) geosite characterization, (d) numerical assessment, and (e) analysis of results [67]. Later on, Brilha (2016) proposed the following stages: (a) a geological/ geomorphological literature review and expert consultation, (b) collating of a list of potential geosites, (c) fieldwork, (d) characterization of geosites, and (e) a quantitative evaluation. After considering these different stages and the actual location, we selected the five steps of the geosite assessment proposed by Brilha in 2016 as cited in [34]. Recently, Mehdioui et al. [68] confirmed that the 'Brilha method' is more suitable and relevant for geotourism development especially for geomorphosite type of assessments.

Thus, the first step in the evaluation consisted of an extensive literature review, including of park management plans, annual reports, and all other existing management reports available to us relating to Yala National Park. Since Yala National Park is open for visitors and a designated protected area under the Fauna and Flora Protection Ordinance, a considerable bulk of secondary literature is available. In addition, we used a Geographic Information System to view the distribution of sites with the help of the Department of Wildlife Conservation, and the Geological Survey and Mines Bureau of Sri Lanka. The second step involved collating a list of potential geosites. This was achieved with the help of key stakeholders and experts at Yala Park including park managers, archaeologists, and geologists as per Hadmoko et al. [69]. During this process, the expert panel selected five potential geosites out of eight. Three sites were removed from the study for several reasons such as the lack of significance of geo-characteristics, geomorphological inaccessibility, risks for both the public and wildlife, and for being isolated from the existing tourist network inside Yala National Park. After selecting five sites, we undertook a primary field survey to familiarise ourselves with the sites and to characterise the sites in accordance with outstanding features such as viewpoints and facilities. This yielded a final list of four potential geosites.

In the final step, we selected the most suitable and reliable quantitative assessment criteria for the evaluation of geosites in the study area. The literature provided a long list of criteria for this purpose, (e.g., [69]). However, we used Sumanapala et al. [70] for guidance who performed an assessment in a similar context, adapting the previous geosite assessment studies by Pereira and Pereira [71], Kubalíková [72], and Kubalíková et al. [73]. Our final list of criteria was classified under the following three themes: scientific value, added value, and tourist value. The individual criteria listed under the three themes were evaluated by an expert panel. The panel consisted of tourism experts, naturalists, and geologists that selected the most suitable criteria out of all. The panel capitalized on their extensive practical knowledge from having worked in the study area.

As a result, we collated a list of 15 criteria under the three themes (Table 2). We used a simple quarter-step scale, as per previous studies [70], to assign a value from 0 to 1 (0 being the lowest) to the 15 criteria for each of the four selected geosites. We did not apply any weights to the criteria as we did not want to introduce an unnecessary bias in this first assessment of geotourism potential. The final step involved scoring using a semi-structured questionnaire [74] to validate or modify the initial site selection, solicitating input from the same experts (park managers, park wardens, archeologists, geologists, and tour operators, total $n = 15$) as in the previous steps.

**Table 2.** Criteria used for the quantitative assessment of the geotourism potential of four sites in Yala National Park, Sri Lanka.

| Theme | Criteria | Definition | Score |
|---|---|---|---|
| Scientific value | Integrity | Generally well-conserved, occasional damage by visitors | 0–1 |
| | Representativeness | Educational importance | 0–1 |
| | Rareness | Possibility to identify exceptional landforms | 0–1 |
| | Geomorphological | Presence of meaningful geomorphology features | 0–1 |
| Added value | Ecological | Presence of protected species | 0–1 |
| | Aesthetical | Aesthetically appealing landscapes | 0–1 |
| | Cultural | Cultural importance | 0–1 |
| | Archaeological | Archaeological importance | 0–1 |
| Tourist value | Protection status | Current protection under government acts | 0–1 |
| | Damage, threats | Uncontrolled visitation | 0–1 |
| | Accessibility | Accessible by suitable transport | 0–1 |
| | Security | Any potential risks or harm to visitors | 0–1 |
| | Site context | Type of landscape | 0–1 |
| | Tourist infrastructure | Paths and structures that facilitate geo-feature observations, and amenities | 0–1 |
| | Educational interest | Visitor interpretation facilities | 0–1 |

To interpret the results, a simple average is used to derive the final score for each criterion, as presented in Table 3. Hence, the total geotourism potential is the sum of the average standard values. In accordance with Kubalíková et al. [75] and Sumanapala et al. [70] we established a threshold that sites needed to reach in the quantitative assessment to be considered for geotourism development: namely, 10 out of 15 points.

**Table 3.** Numerical assessment of geotourism potential of four selected geosites in Yala National Park: Minihagalkanda, Potana, Paranangala, and Elephant Rock. For detailed definitions of criteria see Table 1.

| Criteria | Values * | | | |
|---|---|---|---|---|
| | Minihagalkanda | Potana | Patanangala | Elephant Rock |
| **Scientific value** | | | | |
| Integrity | 1 | 1 | 1 | 0.5 |
| Representativeness | 1 | 0 | 0.75 | 0.5 |
| Rareness | 1 | 1 | 1 | 1 |
| Geomorphological | 1 | 0.5 | 1 | 0.75 |
| **Added value** | | | | |
| Ecological | 1 | 1 | 1 | 1 |
| Aesthetical | 1 | 1 | 1 | 1 |
| Cultural | 0.75 | 0.5 | 1 | 0.75 |
| Archeological | 1 | 1 | 1 | 0.75 |
| **Tourist value** | | | | |
| Protection status | 1 | 1 | 1 | 1 |
| Damage, threats | 0.5 | 0.5 | 0.5 | 0.5 |
| Accessibility | 0.25 | 1 | 0.75 | 0 |
| Security | 1 | 1 | 1 | 1 |
| Site context | 0.5 | 0.5 | 0.5 | 0.5 |
| Tourist infrastructure | 0 | 0 | 0.75 | 0 |
| Educational interest | 1 | 0.75 | 1 | 0 |
| **Total** | **12.25** | **11.50** | **13.25** | **5.5** |

* The evaluation was informed by a review of secondary literature such as park management reports, and through a questionnaire-based survey of park managers, park wardens, archaeologists, geologists, and tour operators at Yala National Park, Sri Lanka.

The quantitative assessment was followed by an analysis of the strengths, weaknesses, opportunities and threats (SWOT) of the four potential geosites, based on a stakeholder questionnaire [70,76]. The SWOT analysis has proved effective in similar studies (e.g., [70,77,78]). It expands the numerical evaluation of sites by providing broader insights into key factors relevant for geotourism such as education options, geotourism practice and conservation measures of proposed geosites inside the park.

## 3. Results and Discussion

The results of the quantitative assessment of the 15 criteria evaluating geotourism potential of the four selected geosites in Yala National Park is presented in Table 3. As for the criteria evaluated under the Scientific value theme, all four sites score highly in terms of rareness (presence of exceptional landforms) which is a key factor to promote for geotourism. However, only the first three sites also managed to preserve the integrity of the attractions. Minihagalkanda and Patanangala stand out among the latter because of their high geomorphological value. Neither Site 1, 2, or 3 show a geomorphology that suggests great educational importance while site 4 reaches a value of 0.5 for its educational importance.

The *Added* value category mainly concerns the ecological, aesthetical, cultural, and archeological value of the different sites. Overall, the added value is highest at Patanangala followed by Minihihagalkanda and Potana. Elephant Rock achieved the lowest marks. From the ecological and aesthetical perspective, all the sites scored equal marks. The first three sites also achieved the highest score for their archaeological value. While Patanangala achieved the highest scores across all Added value criteria, the other sites underperformed for the cultural criterion.

As for the Tourist values, Patanangala scored overall higher marks compared to the other sites, while Elephant Rock scored the lowest overall marks even though it achieved the same scores for security and site context as the other sites.

Consequently, the result of the numerical evaluation points very clearly at three potential geotourism sites for Yala National Park: namely, Patanangala followed by Minihagalkanda and Potana. These three sites achieved an overall score of >10, with Patanangala nearly reaching the perfect score of 15. Elephant Rock was by far outperformed and only reached an overall value of 5.

In conclusion, although Patanangala, already attracts visitors due its aesthetic appeal and the scenic opportunities for photography, offering geotourism activities would add further value and promotional opportunities for the site. It might also attract an additional market segment of visitors whose interest lies specifically in the geomorphological and geological features of the park. Educational opportunities can be explored, for instance, by introducing the formation process of Inselbergs, and by conveying the challenges and threats of these type of formations. This was for instance realized for other limestone Inselbergs such as the famous Twelve Apostles in Port Campbell National Park, Australia, that attract large numbers of visitors. Further value to the tourism experience could come from educating about soil characteristics, coastal landscape diversity and coastal environmental changes due to climate change.

Minihagalkanda is an identified geotourism site. Managers and stakeholders can build on this reputation and develop attractive geotourism experiences here, mainly under three main aspects: geology, geomorphology, and archeology. From a geological point of view, this site should be marketed as one of the best in Sri Lanka to observe the outcrops of the lower shale and limestone bed. The site can also be promoted for its archeological significance and pre-historic human history.

Finally, Potana is still not popular or well known among visitors to Yala National Park. In fact, limited information is available for this site. It does however clearly hold potential for geotourism concerned with the geomorphology of the site. A drawcard here is the relative pristineness of the area which makes it a destination that would appeal to travellers that seek lesser-known destinations.

Following onto the numerical evaluation which indeed identified multiple promising sites for geotourism development, the study conducted a SWOT analysis to determine the general strengths and opportunities for developing geotourism at Yala National Park (Table 4). At the same time, the SWOT analysis helped collating a list of the weaknesses and threats that park management needs to consider and overcome to develop a sustainable form of geotourism that aids in minimizing impacts on overused wildlife.

**Table 4.** SWOT analysis for developing geotourism at Yala National Park, Sri Lanka.

**Strengths**

- Well-known national park in the country.
- Yala has different types of inselbergs, featuring desirable morphological features.
- Protection is high as all sites are managed under the National Park ordinance.
- The ecological value is high because of the mixed dry zone/coastal vegetation habitat.
- Presence of important archeological features of historical value.
- Geological processes and soil formations can be studied which is vital for visitors who are interested in limestone and related things.
- High potential for education.
- Many different activities are possible and can be combined during a park visit (variable "package options").
- Most of the sites are assessable and easy to reach by safari vehicle.
- Some sites (e.g., Patanangala) are already popular among visitors.

**Weaknesses**

- Some sites have restricted access and require permission (e.g., Minihagalkanda), and some are restricted to observation from a safari vehicle.
- Generally, visitor movements in the park are restricted because of the protection afforded to wildlife but also visitors themselves (against dangerous animals).

**Opportunities**

- Targeted promotion of geotourism focussed on the aesthetical and educational value and rareness of features.
- Park managers should increase educational activities around the selected sites and establish supporting educational facilities.
- Park managers can create zones around geosites that people can safely explore afoot.
- Safari operators have an opportunity to add extra activities to their wildlife watching offers.

**Threats**

- Selected sites and surroundings might suffer from visitor impacts.
- Dangerous wildlife may put visitors at risk
- Lack of financial support to maintain geotourism sites.
- Lack of interest in geotourism among the visitors, park operators.

Clearly, Yala National Park harbours great strengths for the development of geotourism opportunities. It is an exceptionally well-known national park in the country that showcases a large variety of geological attractions with added value from its many ecological, aesthetical, cultural and archaeological features [79]. The potential is great to bolster the geotourism offer with high-quality educational programs and material. Previous studies identified interpretation as being very important for nature-based visitors [29], which is critical for developing Yala National Park as a geosite destination [20]. Most of the sites are easily accessible in a reasonable amount of time by safari vehicles, which is a key for visitors travelling through on a tight schedule to other sites outside of the park or for visitors who wish to see other types of attractions in the park. Ginting and Sasmita [80] noted that visitors are expecting support and auxiliary facilities in geotourism destinations, especially in newly developed sites, including viewpoints, directional signage, and an efficient waste management system that reduced littering [81,82]. Scenery, interpretative accessibility, and safety are further considerations [83]. Some of the sites such as Patanangala are already popular among visitors, so word-of-mouth promotion is likely to help with the promotion of this particular site and Yala National Park as a geotourism destination overall.

At the same time, weaknesses were identified: for some sites the identified strengths do not fully apply. For example, Minihagalkanda, because visitation here requires special permission. Thus, even though this site was identified as one with high potential for geotourism, this particular access issue, may prevent further development, unless an easy, fair and sustainable system is implemented to distribute permits while preserving the

integrity of the site. A related issue concerns some of the other sites which can in parts only be viewed from a safari vehicle. Visitors may perceive this as too restrictive. More generally, visitor movements are highly restricted in this park to protect wildlife from visitors but also to protect visitors from dangerous wildlife. Consequently, options to explore the park independently do not exist which may be unattractive to certain segments of the visitor market. This issue needs to be addressed through a risk assessment of the individual sites. Erfurt [84] for example noted that before practicing geotourism activities visitor safety measures and assessments must be implemented along with risk management strategies to ensure visitor safety at the park [85]. Consequently, solutions may be found that harness opportunities; for example, specific zones within geosites could be selected that allow for safe and easy exploration afoot.

Ample opportunities exist to promote geotourism because the sites are attractive and aesthetically appealing, and feature attractions of rare value [86,87]. Interpretation should consist of a multi-tier system including information provided in the Yala National Park visitor centre targeting specific geotourism sites. Geotourists expect attractive interpretation material (e.g., brochures, pocket-sized guidebook, visual material) which attracts visitors and influences their levels of satisfaction [81] and helps them better understand landscape diversity [20]. Next, these sites need to be promoted to multiple stakeholders, such as for instance to safari operators, restaurant and lodge operators, to facilitate word-of-mouth promotion [88,89]; or to measure "Outstanding Universal Values" to convey cultural and natural significance to the overseas tourism market [90].

The intention of our study was to suggest ways to reduce numbers of visitors engaging in wildlife-related activities and observation times. Thus, actions derived from this work should be underpinned by a marketing strategy that affords additional efforts to the promotion of geotourism activities over wildlife tourism activities [91]. As geotourism is not yet properly developed on site, it will benefit from flow-on effects from wildlife tourists. Yet, at the same time increasing promotion of geotourism activities is likely to decrease visitor numbers and time spent with wildlife tourism activities compared to time spent on exploring the geodiversity, culture and history of the park [72,92]. As stated above, the educational value of geotourism is great and the opportunities to develop this aspect of the experience and then use it for promotional purposes are tremendous [93]. Importantly, as tourists tend to evaluate the overall experience [90], geosites that offer reliable observations of attractions will alleviate the pressure for tour operators to 'perform' quite as aggressively when pursuing the rather serendipitous wildlife attractions that visitors may or may not encounter (closely) during their visit. Even a tight coupling of combined wildlife and geotourism experiences could be envisaged. If managed correctly, this harbours great potential as trialled in various other countries such as India, South Africa and Indonesia [90,94,95].

Potential threats that require careful management include human impacts on geosites and their surrounds. While the intention is to reduce impacts on wildlife, at the same time, geosites can also be impacted on [96]. A fine balance needs to be ascertained in visitor numbers and times allocated to geo- and wildlife related activities [97]. Thresholds of usage need to be determined. Thus, decisions need to be made as to which sites to open up for visitors. If park managers are concerned about visitor impacts at geosites in the park, a booking system could be established, especially for Minihagalkanda considering its current access restrictions. We identified three potential sites for geotourism and yet, not all may need to be made accessible. Another general consideration is a potential lack of financial support [98]. Promoting geotourism as an attraction that is not yet established is more costly than promoting wildlife tourism which is already largely self-promoted through word-of-mouth recommendation. In addition, educational facilities, programs and material is costly. Finally, research needs to be conducted to determine whether the demand for geotourism in the area is sufficiently high [95]. The international trend of increasing demand for geotourism certainly suggests that it would be, and yet this needs to be evidenced through more research.

## 4. Conclusions

The novelty of the experience is an important factor for the tourism industry, among other for overcoming the challenges of overuse, overcrowding and related impacts of existing tourism experiences. Numerous studies have therefore been developed to promote geotourism under the banner of sustainability. However, this is the first study to discuss the potential of geotourism as an alternative and diversification strategy to reduce impacts from wildlife tourism in protected areas. We provide a stepwise procedure to identify and numerically assess geosite potential in other protected areas, and to evaluate the strengths, weaknesses, opportunities, and threats of specific geosites. The coupling of the narrative SWOT assessment with the numerically based identification of specific sites serves as a methodological blueprint for other protected areas that intend to diversify their visitor experience offer and aim to capitalize on geotourism as one component in their portfolio.

The core idea of the study is to illustrate the process of identifying geotourism sites to minimize overcrowding and impacts on wildlife tourism attractions which has become a major issue in Yala National Park in Sri Lanka. The intention is to attract visitors to geosites and thereby reduce their numbers and time spent observing wildlife; also in addition, to reduce the intensity by which wildlife is pursued by providing additional reliable attractions that contribute to visitor satisfaction even if wildlife observations are less than optimal. Therefore, this study offers a sustainable solution for overcrowding and overuse of wildlife attractions while highlighting the geological and geomorphological landscape values of a protected area.

We used a numerical evaluation of 15 criteria to identify three geotourism sites. So far, these sites have not been identified as geotourism sites although Patanangala shows the first signs of that. At the same time, the infrastructure at Patanangala currently is only marginally developed. Minihagalkanda and Potana have no visitor facilities at all. To achieve geotourism site status, a strategic development of these sites is necessary, coupled with an intelligent marketing plan underpinned by financial resources that capitalizes on the various added value factors that make a visit to Yala National Park and the potential geotourism sites attractive. Another major concern are the restrictions that impede independent exploration of the park or that offer at least the option to exit safari vehicles in certain demarcated and safe locations. These issues need to be addressed through a careful consideration of zoning and risk management.

Future studies can be focused on developing a weighting system for individual selection criteria for geosites, or on ways to develop promotional and interpretation strategies for geotourism sites. Importantly, future research needs to develop monitoring systems to ensure that geotourism development is managed sustainably.

**Author Contributions:** Conceptualization, D.S. and I.D.W.; methodology, D.S. and I.D.W.; data collection, D.S.: writing—original draft preparation D.S.; writing—review and editing, I.D.W.; supervision, I.D.W.; project administration, D.S. All authors have read and agreed to the published version of the manuscript.

**Funding:** This research received no external funding.

**Data Availability Statement:** Not applicable.

**Acknowledgments:** We thank Pradeep Nalaka Ranasinghe from the University of Ruhuna, Sri Lanka, offered background information. We also want to thank Chandrarathna from the Department of Wildlife Conservation, Sri Lanka and Upul Wickremasinghe offering useful information and providing images for this article.

**Conflicts of Interest:** The authors declare no conflict of interest.

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
