# Peer review of "Introducing Geotourism to Diversify the Visitor Experience in Protected Areas and Reduce Impacts on Overused Attractions"

_land, doi:10.3390/land11122118_

Round 1

Reviewer 1 Report

Geotourism, the form of tourism based on geological resources, has been studied more and more in recent years, and the subject of the article is topical. The area of ​​spread is unlimited,but depends on the factors involved in the development of the region, depending on the geotourism destination.

The current study is focused on four areas that can become important tourist destinations in Sri Lanka. The studied areas have a clear presentation.

The research questions / hypotheses and research method must be clearly described.  

Considering that the purpose of the work is “to diversify the visitor experience”, I consider it necessary the literature in field. Visitors and their experience ensure or not the continuity of the place as a tourist destination. That is why a review of the specialized literature is necessary.

The impact of tourists on the environment must be highlighted in the literature.

Please develop the literature review (last 5 years). I suggest to insert the chapter literature review.

Please insert the source to figures and tables.

Author Response

Dear Reviewer 

Thank you for the opportunity to resubmit a revised version of the manuscript, and to the reviewers, for the valuable feedback to strengthen our manuscript further. we note that the reviewer has all expressed their strong interest in our work. Find the table below our detailed response to the reviewer's comments and remedial actions undertaken.  

Reviewer 2 Report

Dear Author(s),
Thank you for the opportunity to read the paper entitled Introducing ecotourism to diversify the visitor experience in protected areas and reduce impacts on overused resources.

I found this paper very interesting. Great job author(s)! The topic of this paper is interesting, but some improvements would be appreciated. Some parts of the Introduction section belong to the Case Study Area. Research objectives need to be stated clearly. The discussion needs to be expanded and the conclusion needs serious rework.

Title - write "geotourism" instead of "geotourim"

Abstract

Comment 1

Very nicely written abstract. It intrigues and lures readers to read the whole article. Great job author/s!

Introduction

Comment 2

About Sri Lanka, you should move to another section – Case Study Area. It doesn’t belong to the Introduction. (Page 1 – Lines 40-42, Page 2 – Lines 43-65).

Also about geosites. Only mention in the Introduction, but the description belongs to Case Study Area.

Comment 3

I suggest making a new paragraph that describes the research objectives.

Comment 4

At the end of the section, you can mention how sections in the manuscript have been organized.

Methods

Comment 5

Subsection – Change the name to the Case Study Area and move paragraphs from the Introduction.

Comment 6

Line 139 - However this is not a criterion that war required in the context of Yala National Park as it was already established that – “was” required. Please modify this.

Results and Discussion

Comment 7

You should expand the discussion. Compare this to the previous studies and references you used in the introduction section.

Conclusion

Comment 8

Explain the importance of your work, novelty, and originality, what are the most relevant conclusions you can make based on the article? What are the limitations of your article? Future research? Theoretical contribution? These segments are missing. The article is very interesting, but I think you can conclude it much better.

Once again thank you very much for the opportunity to read this interesting article. The manuscript is really nicely written with interesting results. Congratulations on the great work author/s! But there are some things that should be improved.  Looking forward to reading your article again.

Wish you all the best!

Sincerely,

Reviewer 

Author Response

Dear Reviewer 

Thank you for the opportunity to resubmit a revised version of the manuscript, and to the reviewers, for the valuable feedback to strengthen our manuscript further. We note that the reviewer has all expressed their strong interest in our work. Please find in the table below our detailed response to the reviewer's comments and remedial actions undertaken. 

Reviewer 3 Report

The article should be improved significantly in order to be published. The main concerns are:

-  Extensive editing of English language and style required. In several sections, it's very hard to understand the paper since words are not used in the proper context eg: "However this is not a criterion that war required in the context of Yala National Park".

- The literature review should be extended especially in the field of the research:  "quantitative assessment of the geotourism potential". The description of the national parks it is not literature review. The literature review should take the readers of the paper towards the methodology and present the main opinions in the field.

- The methodology, as well, is extremely simple and irrelevant to the solve the proposed research questions. A proper and more relevant methodology should be proposed, especially after studying more the literature in the field and other case studies. Not all the elements should have the same weight in the study.  

- It is not normal to take sentences from other authors and change only one word e.g.: "Geotourism is currently one of the fastest developing branches of tourism world wide with huge untapped potential". The only word which is changed is "developing" instead of the original growing. This situation cab be considered plagiarism. The article contains several sentences like the highlighted paragraph.

- The article should be seriously reviewed in order to increase its scientific soundness.

Author Response

Dear Reviewer 

Thank you for the opportunity to resubmit a revised version of the manuscript, and to the reviewers, for the valuable feedback to strengthen our manuscript further. We note that the reviewer has expressed strong interest in our work. Please find in the table below our detailed response to the reviewer's comments and remedial actions undertaken.    

Reviewer 4 Report

I carefully read the present version of the ms "Introducing geotourism to diversify the visitor experience in protected areas and reduce impacts on overused resources". I think that the idea of popularizing geological features in Sri Lanka in areas which are overused by tourists interested in wildlife is valid and deservs to be taken into account. However, I think the ms in its present form suffers some problems. What I think is that it is not a matter of science, but a matter of presentation. First of all, the ms reading leaves you with the impression that it has been written by two different hands. In fact, results and  the first part of the discussion session are quite rough and seem somewhat superficial, and this is very far from giving a good impression to the reader. Both expert elicitation and quality evaluation tables shoud be better refines as topresentation. The same think I woud say about photo and figures. These authors are dealing with geoscience popularization, why do not add, as an example, a sketch for inselberg formation? Why only three photos whose quality is very low? The final part of the discussion seems much better, since it is well structured and the various statements are fully sound and well argumented. Anyway, I attach an annotated .pdf with some suggestions of mine.

Author Response

Dear Reviewer 

Thank you for the opportunity to resubmit a revised version of the manuscript, and to the reviewer, for the valuable feedback to strengthen our manuscript further. We note that the reviewer has expressed a strong interest in our work. Please find the table below for our detailed response to the reviewer's comments and remedial actions undertaken.  

Round 2

Reviewer 1 Report

Dear Author, 

 I had the pleasure to review your paper, an important subject to study. Thank you for following the recommendation. Succes! 

Reviewer 4 Report

I think the authors have made a lot of effort to improve the quality of presentation. I still think that the almost  total absence of high-quality images and drawings lowers the quality of the result, but the authors wrote that most photos are copyrighted...